# Adaptive Incentive Design for Markov Decision Processes with Unknown Rewards*

**Haoxiang Ma**                                                    *hma2@ufl.edu*
*Department of Electrical and Computer Engineering*
*University of Florida*

**Shuo Han**                                                       *hanshuo@uic.edu*
*Department of Electrical and Computer Engineering*
*University of Illinois Chicago*

**Ahmed Hemida**                                    *ahmed.h.hemida.ctr@army.mil*
*DEVCOM Army Research Laboratory*

**Charles Kamhoua**                                *charles.a.kamhoua.civ@army.mil*
*DEVCOM Army Research Laboratory*

**Jie Fu**                                                          *fujie@ufl.edu*
*Department of Electrical and Computer Engineering*
*University of Florida*

## Abstract

Incentive design, also known as model design or environment design for Markov decision processes(MDPs), refers to a class of problems in which a leader can incentivize his follower by modifying the follower's reward function, in anticipation that the follower's optimal policy in the resulting MDP can be desirable for the leader's objective. In this work, we propose gradient-ascent algorithms to compute the leader's optimal incentive design, despite the lack of knowledge about the follower's reward function. First, we formulate the incentive design problem as a bi-level optimization problem and demonstrate that, by the softmax temporal consistency between the follower's policy and value function, the bi-level optimization problem can be reduced to single-level optimization, for which a gradient-based algorithm can be developed to optimize the leader's objective. We establish several key properties of incentive design in MDPs and prove the convergence of the proposed gradient-based method. Next, we show that the gradient terms can be estimated from observations of the follower's best response policy, enabling the use of a stochastic gradient-ascent algorithm to compute a locally optimal incentive design without knowing or learning the follower's reward function. Finally, we analyze the conditions under which an incentive design remains optimal for two different rewards which are policy invariant. The effectiveness of the proposed algorithm is demonstrated using a small probabilistic transition system and a stochastic gridworld.

## 1 Introduction

Incentive design, also known as the *principal-agent* problem (Bolton & Dewatripont, 2005), is concerned with applications where a leader aims to optimize the performance of a system which multiple people/ agents actively interact with. Incentive design has been widely explored in economics (Seabright, 1993; Kamenica, 2012; Easley & Ghosh, 2016; Athey & Roberts, 2001) and control systems (Ho et al., 1981; Ho & Teneketzis, 1984; Ratliff & Fiez, 2020), with applications such as demand response (Zhou et al., 2017; Dobakhshari & Gupta, 2016) and network congestion control (Barrera & Garcia, 2014; Alpcan et al., 2009; Li et al., 2017).

---

*"Distribution A: Approved for Public Release, Distribution is Unlimited."

From a game-theoretic perspective, incentive design can be viewed as a non-cooperative Stackelberg game (Simaan & Cruz Jr, 1973), also called a leader-follower game. In a leader-follower game, the follower faces a planning problem that can be influenced by an incentive policy designed by the leader. Once the incentive policy is chosen, the follower will play a best response policy that optimizes the objective of the planning problem. With the anticipation of the follower's best response policy, the leader aims to choose the incentive policy to align the follower's policy with the objective of the leader, subject to limited resources to realize the incentive policy.

In this work, we consider a special class of leader-follower games: The follower faces a sequential planning problem described by a Markov decision process (MDP), whereas the incentive policy of the leader can be used to provide additional side-payments or rewards to the follower at the cost of the leader's own payoff. The leader has complete knowledge of the MDP faced by the follower except the follower's reward function. The leader interacts with the follower repeatedly, during which the leader can observe the follower's decisions and tunes the incentive policy based on the observations.

Our setting of an unknown follower's reward is different from most existing settings in incentive design, which assume that the leader knows the follower's reward function. Relaxing this assumption is crucial for many practical applications: In congestion control, a traffic management system cannot know individual drivers' preferred routes and destinations; in resource allocation for defense in security, a defender in general has limited or incomplete information about the attacker's goal or intention. These applications motivate the following question: *Despite the unknown reward of the follower, can we learn an optimal incentive policy when it is possible to interact with the follower multiple times?* To this end, we develop a gradient-based optimization method that computes a locally optimal incentive policy and prove its convergence. We also develop a method to compute a unbiased estimate of the gradient, based on the observed trajectories from the follower's best-response policy. The gradient estimation leads to an adaptive incentive design method for the unknown follower's reward case.

**Related work** Incentive design where the follower's planning problem is modeled as an MDP has been studied under several other names including "reward tempering" (Everitt et al., 2021; Skalse et al., 2022), "model design" (Chen et al., 2022), "behavior modification" (Savas et al., 2022), and "environment design" (Zhang & Parkes, 2008; Zhang et al., 2009). Zhang & Parkes (2008) first introduced environment design in the MDP setting and developed a mixed-integer programming approach to solve the optimal design assuming the follower best responds to the designed MDP with a deterministic optimal policy. Zhang et al. (2009) further generalized the environment design and provided a binary-search algorithm with logarithmic convergence for the case when an agent's model parameters are initially unknown to the designer.

In (Chakraborty et al., 2023), the authors study a similar bi-level optimization formulation to align the follower's policy with the leader's interests, where the leader is allowed to directly choose the parameters that define the follower's reward function. In order to compute the gradient of the follower's policy mapping with respect to the reward parameters, both the Jacobian and the Hessian of the lower-level objective (i.e., the follower's value function) have to be evaluated. In (Chen et al., 2022), the authors propose a gradient-based algorithm to regulate the follower's reward function or transition dynamics by selecting a design parameter. The gradient computation requires evaluating the gradients of the value function, the Q-function, and the advantage function in the design parameter. They developed a provably convergent single-loop gradient algorithm to update the MDP policy and the design parameter. The authors mentioned that in the case when the pre-regulated MDP is unknown, one can learn the transition model and the reward function using offline data. Our gradient-based optimal incentive design does not require learning the follower's reward function— the gradient can be computed using the sampled trajectories under the follower's best response policy.

Savas et al. (2019) studied *sequential incentive* design for a principal agent to optimize the probability of satisfying a temporal logic formula. Sequential incentive design differs from environment design or model design as the principle can provide incentive dynamically based on the agent's current state , rather than static design, anticipating the agent's best response. As a result, sequential incentive design can be formulated as an MDP with an augmented state space. Savas et al. (2022) studied the sequential incentive design in a setup with a myopic follower who selects the optimal action based on the sum of the current reward

and the provided incentive from the leader. They showed that the sequential incentive design problem is PSPACE-hard but computing an $\epsilon$-optimal solution can be reformulated as a constrained MDP.

However, all these aforementioned works assume that the follower's reward function is known to the leader. Ratliff & Fiez (2020) studied adaptive incentive problem with multiple followers, whose decision-making processes are unknown to the leader. They consider another class of incentive design where the follower's planning problem is a generalized linear model, rather than an MDP with modifiable reward function. Another related line of work is optimization with decision-dependent distribution (Wood & Dall'Anese, 2023; Drusvyatskiy & Xiao, 2023), where the variability in the follower's policies is modeled as a distribution that depends on the leader's decision variable. However, the optimization method does not adjust the leader's decision based on the observation of the follower's responses.

**Our contributions** Compared to existing work, our work has the following contributions:

1. The aforementioned work on incentive design in MDPs requires knowing the follower's reward function, either by having it as prior knowledge or learning the follower's reward from observed behaviors. Our approach does not rely on knowing the follower's reward function and only assumes that the follower's response satisfies softmax temporal consistency (Nachum et al., 2017) and that the follower uses a softmax policy. The resulting incentivizing strategy for the leader is an adaptive law that chooses the incentivizing decisions based the historical response from the follower.

2. We propose a gradient-based method to compute the optimal incentive for the follower. Leveraging the softmax parameterization and the property of entropy-regularized optimal policies, we develop a method to compute the exact gradient of the follower's policy with respect to the incentive design variables. The method involves multiple policy evaluations, each for a different "reward" function (see Section 3). As a main advantage, our method avoids the computationally expensive procedure of evaluating the Jacobian and the Hessian of the lower-level objective, which are used in hypergradient descent methods in bi-level optimization . Further, our method leads to adaptive incentive design using estimated gradients. We formally prove that the proposed method is guaranteed to converge to a locally optimal incentive policy.

3. Building on the insight from the gradient-based method, we develop an unbiased estimator of the gradient terms, which enables us to extend the method to the case when the follower's reward function is unknown. Our method does not require reward learning, which differs from other adaptive model design method (Chen et al., 2022).

4. Lastly, we prove that under certain conditions, the optimal incentive design for a follower with reward function $R$ remains to be optimal for a follower with reward function $R^{\dagger}$, provided that $R$ and $R^{\dagger}$ are policy-invariant under reward shaping. This result has two implications: 1) It shows that our method for the reward-known case is also applicable to the case when the follower's reward function is unknown but learned from data, provided that the learned reward $R^{\dagger}$ is policy-invariant to the true reward $R$ under reward shaping; 2) The leader only needs to design one incentive policy to a set of followers whose different reward functions are policy-invariant.

## 2 Preliminaries and Problem Formulation

**Notations** Let $\mathbf{R}$ denote the set of real numbers and $\mathbf{R}^n$ the set of real $n$-vectors. The vector of all ones is represented as $\mathbf{1}$ with the dimension understood from the context. The notation $z_i$ refers to the $i$-th component of a vector $z \in \mathbf{R}^n$ or to the $i$-th element of a sequence $z_1, z_2, \ldots$, which will be clarified by the context. The set of probability distributions over a finite set $Z$ is denoted as $\mathcal{D}(Z)$.

We consider an incentive design problem where an agent (the leader, player 1/P1) can incentivize his opponent (the follower, player 2/P2) with a side payment, in anticipation that the best response of P2, given the combined side payment and P2's original reward, can maximize P1's utility.

We start by modeling the interaction dynamics between P2 and a stochastic environment as an MDP:

$$M = (S, A, P, \mu),$$

where $S$ is a set of states, $A$ is a set of actions, $P : S \times A \to \mathcal{D}(S)$ is a probabilistic transition function such that $P(s'|s, a)$ is the probability of reaching state $s'$ given action $a$ being taken in state $s$, and $\mu \in \mathcal{D}(S)$ is the initial state distribution. Throughout the paper, we assume that both $S$ and $A$ are finite. Under this assumption, for any function $f : S \times A \to \mathbf{R}$, we sometimes view $f$ as a vector in $\mathbf{R}^{|S \times A|}$ and use $f_{s,a}$ in place of $f(s, a)$.

**The leader (P1)'s objective**   P1's objective is given by a reward function $R_1 : S \times A \to \mathbf{R}$. However, P1 cannot take actions in the MDP but relies on P2's policy to obtain reward. Given a P2's policy $\pi$ and the initial state distribution $\mu$, the total discounted reward received by P1 is

$$V_1(\mu, \pi) = \mathbb{E}_\pi \left[ \sum_{t=0}^{\infty} \gamma^t R_1(S_t, A_t) \mid S_0 \sim \mu \right].$$

where $\gamma$ is a discount factor. That is, P1's value is defined by evaluating P2's policy in the MDP $M$ given P1's reward function.

Because P1's objective depends on P2's policy, P1 aims to incentivize P2 to take a policy that maximizes P1's own value.

**Incentives as side payments**   In the MDP, P2's original reward function without any incentive is $\bar{R}_2 : S \times A \to \mathbf{R}$ where $\bar{R}_2(s, a)$ is the reward received by P2 for taking action $a$ in state $s$. P1's incentive to P2 is represented as a function $x : S \times A \to \mathbf{R}_+$, hereafter referred to as the *side payment*. Specifically, $x(s, a)$ is the additional non-negative reward that P1 offers to P2 when P2 takes action $a$ in state $s$.

Given a side payment $x$, P2's modified reward function $R_2(x)$ is defined as follows: For all $(s, a) \in S \times A$,

$$R_2(s, a; x) = \bar{R}_2(s, a) + x(s, a). \tag{1}$$

We model P2's planning problem with side payment $x$ as an entropy-regularized MDP

$$M(x) = (S, A, P, \mu, \gamma, R_2(x)).$$

The value function of the entropy-regularized MDP is defined by

$$V_2(s, R_2(x), \pi) = \mathbb{E}_\pi \left[ \sum_{t=0}^{\infty} \gamma^t R_2(S_t, A_t; x) - \tau \log \pi(S_t, A_t) \,\middle|\, S_0 = s \right],$$

where $\tau > 0$ is the temperature parameter that controls the amount of entropy regularization and reflects the level of rationality of P2. We assume that the temperature parameter is common knowledge. The value given the initial distribution $\mu$ is written as $V_2(\mu, R_2(x), \pi) = \mathbb{E}_{s \sim \mu} V_2(s, R_2(x), \pi)$.

**Problem 1** (Incentive design). *P1's incentive design problem is the following bi-level optimization problem:*

$$
\begin{aligned}
\underset{x \in \mathbf{R}_+^{|S \times A|}, \ \pi^\star}{\text{maximize}} \quad & V_1(\mu, \pi^\star) - h(x) \\
\text{subject to} \quad & \pi^\star \in \underset{\pi \in \Pi}{\arg \max} \, V_2(\mu, R_2(x), \pi),
\end{aligned}
\tag{2}
$$

*where $h$ is a cost function for side payment. The function $h$ is $L_h$-Lipschitz continuous and $L$-smooth (i.e., $\nabla h$ is $L$-Lipschitz continuous).*

One particular choice of $h$ is given by $h(x) = c\|x\|_1$ for some constant $c > 0$.

**Remark 1.** *P2's reward function with the incentive can be more general than the sum of P2's original reward and the side payment. Our results only require knowing the derivative of $R_2(s, a; \cdot)$ for all $s \in S$ and $a \in A$. This holds, for example, when $R_2(s, a; x) = \bar{R}_2(s, a) + f(x, s, a)$, where $f(\cdot, s, a)$ is an $L_f$-Lipschitz-continuous function for all $s \in S$ and $a \in A$. We omit this generalization for clarity in the main results.*

The optimal value function $V_2^\star$ of the entropy-regularized MDP satisfies the following entropy-regularized Bellman equation (Nachum et al., 2017):

$$V_2^\star(s, R_2(x)) = \tau \log \sum_{a \in A} \exp\{(R_2(s, a; x) + \gamma \mathbb{E}_{s' \sim P(\cdot|s,a)} V_2^\star(s', R_2(x)))/\tau\}, \quad \forall s \in S. \tag{3}$$

Note that, as $\tau$ approaches 0, equation 3 recovers the standard optimal Bellman equation.

Let $Q^\star(R_2(x)) \colon S \times A \to \mathbf{R}$ be the optimal state-action value function of the entropy-regularized MDP under reward $R_2(x)$:

$$Q^\star(s, a, R_2(x)) = R_2(s, a; x) + \mathbb{E}_{s' \sim P(\cdot|s,a)} V_2^\star(s', R_2(x)).$$

For a fixed temperature parameter $\tau$, the optimal policy of the entropy-regularized MDP is uniquely defined by

$$\pi^\star(s, a) = \frac{\exp(Q^\star(s, a, R_2(x))/\tau)}{\sum_{a' \in A} \exp(Q^\star(s, a', R_2(x))/\tau)}. \tag{4}$$

For convenience, for any $\theta \in \mathbf{R}^{|S \times A|}$, define the *softmax policy* $\pi_\theta$ as

$$\pi_\theta(s, a) = \frac{\exp(\theta_{s,a}/\tau)}{\sum_{a' \in A} \exp(\theta_{s,a'}/\tau)}. \tag{5}$$

Then, the optimal policy of the entropy-regularized MDP can be written succinctly as $\pi_{Q^\star(R_2(x))}$, where $Q^\star(R_2(x))$ is viewed as a vector in $\mathbf{R}^{|S \times A|}$.

## 3   Adaptive Incentive Design via Gradient Ascent

Due to the relation between the optimal policy and the optimal state-action value function of the entropy-regularized MDP given by equation 4, the lower-level problem in the bilevel optimization problem in equation 2 has a unique solution $\pi_{Q^\star(R_2(x))}$. Thus, the bi-level optimization problem reduces to a single-level optimization problem using constraint elimination:

$$\underset{x \in \mathbf{R}_+^{|S \times A|}}{\text{maximize}} \quad V_1(\mu, \pi_{Q^\star(R_2(x))}) - h(x). \tag{6}$$

For convenience, we denote the objective function by $J(x) \triangleq V_1(\mu, \pi_{Q^\star(R_2(x))}) - h(x)$.

**Lemma 1.** *The function $J(x)$ can be non-concave.*

*Proof.* We provide a concrete example, in which the objective function $J(x)$ is non-concave. Consider a simple MDP in Figure 1a, both states 1 and 2 are sink states. The follower receives a reward of $R_2(1) = 0$ if state 1 is reached, and $R_2(2) = 1$ if state 2 is reached. The leader receives a reward of $R_1(1) = 1$ when state 1 is reached, and $R_1(2) = 0$ if state 2 is reached. The temperature parameter $\tau = 0.1$. The leader can only assign a nonnegative sidepayment $x$ to state 1. The cost function $h(x) = |x|$.

The follower has two actions $a_1, a_2$ from state 0 and both have deterministic outcomes to reach state 1 and 2, respectively.

In this example, the follower's best response policy given leader's decision $x$ is $\pi(0, a_1; x) \propto \exp((R_2(1)+x)/\tau)$ and $\pi(0, a_2; x) \propto \exp((R_2(2) + x)/\tau)$.

The objective function is

$$J(x) = R_1(1) \cdot \pi(0, a_1) + R_1(2) \cdot \pi(0, a_2) - h(x).$$

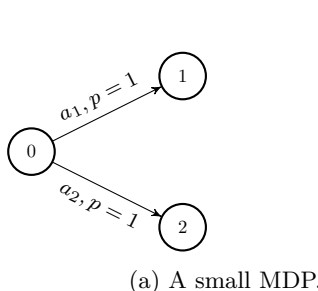

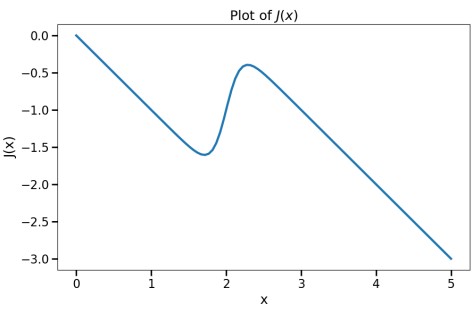

(a) A small MDP.  (b) The plot of the objective function $J(x)$.

Figure 1: An example to illustrate the non-concavity of $J(x)$.

The function $J(x)$ for $x \in [0, \infty)$ has two local maximal solutions and one global optimal at $x = 0$, as shown in Fig. 1b.

$\square$

### 3.1 Computing the total gradient: The case with known P2's reward

We consider using the gradient-ascent method to find $x$ that maximizes P1's objective function. For any $\theta$, define $J_1(\theta) \triangleq V_1(\mu, \pi_\theta)$, using which one can write $J(x) = J_1(Q^\star(R_2(x))) - h(x)$. Following the chain rule, the derivative of $J$ is given by

$$DJ(x) = DJ_1(Q^\star(R_2(x))) \cdot DQ^\star(R_2(x)) \cdot DR_2(x) - Dh(x). \tag{7}$$

Because $h$ is given, $Dh(x)$ can be computed analytically. Similarly, $DR_2(x)$ can be computed analytically given the function $R_2$. The derivative $DJ_1$ can be computed using the following proposition. (Recall that the derivative and the gradient of a real-valued function $J_1$ are related by $DJ_1(\theta) = \nabla J_1(\theta)^T$.)

**Proposition 1** (Agarwal et al. (2021)). *Let $\pi_\theta$ be the softmax policy defined by equation 5. Then*

$$\nabla J_1(\theta) = \nabla \mathbb{E}_{\pi_\theta}[R_1(\rho)] = \mathbb{E}_{\pi_\theta}\left[R_1(\rho)\nabla \log \pi_\theta(\rho)\right], \tag{8}$$

*where $\rho = s_0, a_0, s_1, a_1, \ldots s_n, a_n$ is the path generated from the Markov chain $M_{\pi_\theta}$, $\pi_\theta(\rho) = \prod_{t=0}^{n} P(s_{t+1}|s_t, a_t)\pi_\theta(s_t, a_t)$ is the probability of the path $\rho$ from the Markov chain $M_{\pi_\theta}$, and $R_1(\rho) = \sum_{k=0}^{n} \gamma^k R_1(s_k, a_k)$ is the cumulative reward of the path $\rho$. For a path $\rho$, the gradient $\nabla \log \pi_\theta(\rho) = \sum_{k=0}^{n} \nabla \log \pi_\theta(s_k, a_k)$, where $\nabla \log \pi_\theta(s, a)$ can be computed as follows:*

$$\frac{\partial \log \pi_\theta(s,a)}{\partial \theta_{\tilde{s},\tilde{a}}} = \begin{cases} 0 & \text{if } s \neq \tilde{s}, \\ (1 - \pi_\theta(s,a))/\tau & \text{if } s = \tilde{s} \text{ and } a = \tilde{a}, \\ -\pi_\theta(s,\tilde{a})/\tau & \text{if } s = \tilde{s} \text{ and } a \neq \tilde{a}. \end{cases} \tag{9}$$

The derivative $DQ^\star$ can be computed using the following proposition.

**Proposition 2.** *Consider an infinite-horizon MDP $M = (S, A, P, s_0, \gamma, r)$ with discounting. Let $Q^\star(r) \colon S \times A \to \mathbf{R}$ be the optimal state-action value function of the entropy-regularized MDP under the reward function $r$. For any $(s, a), (\tilde{s}, \tilde{a}) \in S \times A$, it holds that*

$$\frac{\partial Q^\star_{s,a}}{\partial r_{\tilde{s},\tilde{a}}} = \mathbf{1}_{(\tilde{s},\tilde{a})}(s,a) + \gamma \mathbb{E}_{s' \sim P(\cdot|s,a)} \sum_{a' \in A} \pi_{Q^\star(r)}(s',a')\frac{\partial Q^\star_{s',a'}}{\partial r_{\tilde{s},\tilde{a}}}, \tag{10}$$

*where*

$$\mathbf{1}_{(\tilde{s},\tilde{a})}(s,a) = \begin{cases} 1 & \text{if } (s,a) = (\tilde{s}, \tilde{a}), \\ 0 & \text{otherwise.} \end{cases} \tag{11}$$

*Proof.* According to the entropy-regularized Bellman equation (Nachum et al., 2017), the optimal state-action value function $Q^\star$ satisfies

$$Q^\star(s,a,r) = r(s,a) + \gamma \mathbb{E}_{s'\sim P(\cdot|s,a)}\left[\tau \log \sum_{a'\in A} \exp\left(Q^\star(s',a',r)/\tau\right)\right],$$

or equivalently, by treating both $Q^\star(r)$ and $r$ as vectors in $\mathbf{R}^{|S\times A|}$,

$$Q^\star_{s,a}(r) = r_{s,a} + \gamma \mathbb{E}_{s'\sim P(\cdot|s,a)}\left[\tau \log \sum_{a'\in A} \exp\left(Q^\star_{s',a'}(r)/\tau\right)\right]. \tag{12}$$

Let $Z(s) \triangleq \sum_{a\in\mathcal{A}} \exp\left(Q^\star(s,a,r)/\tau\right) = \sum_{a\in\mathcal{A}} \exp\left(Q^\star_{s,a}(r)/\tau\right)$. Taking the derivative with respect to $r_{\tilde{s},\tilde{a}}$ on both sides of equation 12, one can obtain

$$\frac{\partial Q^\star_{s,a}}{\partial r_{\tilde{s},\tilde{a}}} = \mathbf{1}_{(\tilde{s},\tilde{a})}(s,a) + \gamma \mathbb{E}_{s'\sim P(\cdot|s,a)}\left[\tau \frac{1}{Z(s')} \sum_{a'\in\mathcal{A}} \frac{1}{\tau} \frac{\partial Q^\star_{s',a'}}{\partial r_{\tilde{s},\tilde{a}}} \exp(Q^\star_{s',a'}(r)/\tau)\right]$$

$$= \mathbf{1}_{(\tilde{s},\tilde{a})}(s,a) + \gamma \mathbb{E}_{s'\sim P(\cdot|s,a)}\left[\frac{1}{Z(s')} \sum_{a'\in\mathcal{A}} \frac{\partial Q^\star_{s',a'}}{\partial r_{\tilde{s},\tilde{a}}} \exp(Q^\star_{s',a'}(r)/\tau)\right]$$

$$= \mathbf{1}_{(\tilde{s},\tilde{a})}(s,a) + \gamma \mathbb{E}_{s'\sim P(\cdot|s,a)} \sum_{a'\in A} \pi_{Q^\star(r)}(s',a') \frac{\partial Q^\star_{s',a'}}{\partial r_{\tilde{s},\tilde{a}}},$$

where the last step follows from $\pi_{Q^\star(r)}(s',a') = \exp(Q^\star(s',a',r)/\tau)/Z(s') = \exp(Q^\star_{s',a'}(r)/\tau)/Z(s')$. $\qquad\square$

For any given $(\tilde{s},\tilde{a}) \in S \times A$, equation 10 is in the form of the standard Bellman equation for state-action value functions. This implies that the partial derivatives

$$\left\{\frac{\partial Q^\star_{s,a}}{\partial r_{\tilde{s},\tilde{a}}} \,\middle|\, s \in S,\ a \in A\right\}$$

can be computed using any method for solving the Bellman equation. The partial derivative of $Q^\star_{s,a}$ with respect to the reward $r_{\tilde{s},\tilde{a}}$ can be interpreted as the discounted number of visiting state-action pair $(\tilde{s},\tilde{a})$ under the entropy-regularized optimal policy $\pi_{Q^\star(r)}$ after taking the action $a$ in the state $s$. If a state-action pair is visited more frequently, then the optimal state-action value function is more sensitive to the changes in the reward for the state-action pair. Moreover, the derivative with respect to the current state-action pair is also influenced by the derivative with respect to future state-action pairs under $\pi_{Q^\star(r)}$.

Algorithm 1 describes the gradient ascent method.

Next, we will conduct the convergence analysis of our algorithm given known P2's reward.

**Proposition 3.** *The total derivative $DJ$ of the objective function $J$ is Lipschitz continuous: There exists $L > 0$ such that for all $x_1, x_2$,*
$$\|DJ(x_1) - DJ(x_2)\| \le L\|x_1 - x_2\|.$$

*Proof.* See Appendix A.2. $\qquad\square$

**Theorem 1.** *When $\eta \le 1/L$, Algorithm 1 converges asymptotically to a stationary point of $J$ as $k \to \infty$.*

**Complexity analysis for per-step gradient computation** For each state-action pair $(s,a)$, the complexity to calculate $\frac{\partial Q^\star}{\partial r_{s,a}}$ is $\mathcal{O}(|S|^2|A|)$ because it is equivalent to policy evaluation. Consider the reward function with side payment defined as a linear combination of P2's original reward and side payment, $R_2(s,a,x) = \bar{R}_2(s,a) + x(s,a)$, for each $(s,a) \in S \times A$. Suppose that only a subset $Z \subseteq S \times A$ of state-action pairs are allocated with nonzero side payments, that is, $x(s,a) \ge 0$ for $(s,a) \in Z$ and $x(s,a) = 0$ for $(s,a) \notin Z$, then the time complexity to calculate $DQ^\star(R_2(x)) \cdot DR_2(x)$ is $\mathcal{O}(|Z| \cdot |S|^2|A|)$. This analysis

**Algorithm 1** Incentive design given P2's reward

---
1: **procedure** GRADIENT-BASED INCENTIVE DESIGN GIVEN P2'S REWARD
2:     Initialize $x_0, \eta, \delta_0 > 0$;         ▷ $\eta$ is the step size, $\delta_0$ is a termination threshold and is close to 0.
3:     $k \leftarrow 0$;
4:     Solve P2's optimal state-action value function of $Q^\star(R_2(x_0))$;
5:     $J_1^0 \leftarrow J(x_0)$;
6:     $\delta \leftarrow \infty$;
7:     **while** $\delta \geq \delta_0$ **do**
8:         Calculate the gradient $DJ(x_k)$;
9:         $x_{k+1} \leftarrow \mathsf{Proj}_{\mathbf{R}_+^{|S \times A|}}(x_k + \eta DJ(x_k))$;         ▷ Projection to non-negative orthant.
10:        $J^{k+1} \leftarrow J(x_{k+1})$;
11:        $\delta \leftarrow |J^{k+1} - J^k|$;
12:        $k \leftarrow k + 1$;
13:     **end while**
14:     **return** $x_k$.
15: **end procedure**

---

informs that in practice, the leader may need to constrain the number of state-action pairs to be allocated with side payment to reduce the complexity in calculating the gradient terms, instead of allocating side payment to every state-action pair.

In the general case of reward with side payment, for each $(s,a) \in S \times A$, $R_2(s,a,x) = \bar{R}_2(s,a) + f(x,s,a)$. Let $x \in \mathbf{R}_+^K$ be a $K$-dimensional vector. Thus, the term $DR_2(x)$ is a dense square matrix of size $(|S||A| \times K)$. The term $DQ^\star(R_2(x))$ is a dense square matrix of size $(|S||A|)^2$ and is computed with a time complexity of $\mathcal{O}(|S|^2|A| \times |S||A|)$ because $\frac{\partial Q^\star}{\partial r_{s,a}}$ is to be computed for each $(s,a) \in S \times A$. Thus, the time complexity of calculating $DQ^\star(R_2(x)) \cdot DR_2(x)$ in the general case is given by $\mathcal{O}(K|S|^2|A|^2 + |S|^3|A|^2)$.

### 3.2   Estimating the total gradient: The case with unknown P2's reward

In the previous section, we discuss how to use a gradient-based algorithm to find an optimal side payment that maximizes P1's total payoff. However, the method assumes that P2's reward function is known to P1. In this section, we show that the proposed gradient-based algorithm can be extended to the design of adaptive incentive when P2's reward is unknown. In this setting, at the beginning of the interaction, P1 has incomplete information about P2's reward function $R_2$. Instead, P1 can observe P2's decisions (in the form of state-action trajectories) under any side payment chosen by P1. P1 is allowed to interact with P2 repeatedly by choosing different side payments, through which P1 aims to eventually find a near-optimal side payment. The term *adaptive incentive design* is analogous to adaptive control, because the algorithm adapts to the unknown reward of P2 and optimizes the side payment through interactions.

Our approach is to estimate the derivative $DJ$ using the trajectories generated by P2. First, it can be seen from equation 7 that both $DR_2(x)$ and $Dh(x)$ are independent of P2's reward function and can be computed as before. For $DJ_1(Q^\star(R_2(x)))$, recall the following sample-based approximation of $DJ_1$:

$$DJ_1(\theta) = \mathbb{E}_{\pi_\theta}\left[R_1(\rho)\nabla \log \pi_\theta(\rho)\right] \approx \frac{1}{|X|}\sum_{i=1}^N R_1(\rho_i)\nabla \log \pi_\theta(\rho_i) = \frac{1}{|X|}\sum_{i=1}^N R_1(\rho_i) \sum_{(s,a) \in \rho_i} \nabla \log \pi_\theta(s,a), \quad (13)$$

where $X = \{\rho_i \mid i = 1,\ldots,N\}$ is a set of sampled trajectories under $\pi_\theta$ for $\theta = Q^\star(R_2(x))$. However, because P1 has no information about P2's original reward $R_2$, given a side payment $x$, P1 cannot solve P2's entropy-regularized optimal policy and thus cannot directly compute the gradient $\nabla \log \pi_\theta(\rho)$.

In order to calculate the gradient term in equation 13 without knowing the exact policy, we estimate a policy from the sampled trajectories. If a state-action pair $(s,a)$ appears in $\rho_i \in X$ for some $i$, then we need to compute $\nabla \log \pi_\theta(s,a)$. A simple maximum-likelihood estimator for $\pi_\theta$ is $\hat{\pi}_\theta(s,a) = \frac{\mathbb{N}(s,a)}{\mathbb{N}(s)}$, where $\mathbb{N}(s,a)$ is

the number of times that $(s,a)$ appears in $X$ and $\mathbb{N}(s) \triangleq \sum_{a \in A} \mathbb{N}(s,a)$. The estimated policy $\hat{\pi}_\theta$ can be used to approximate $\nabla \log \pi_\theta(s,a)$ based on equation 9: $\frac{\partial \log \pi_\theta(s,a)}{\partial \theta_{s,a}} = \frac{1}{\tau}(1 - \pi_\theta(s,a)) \approx \frac{1}{\tau}(1 - \hat{\pi}_\theta(s,a))$, and $\frac{\partial \log \pi_\theta(s,a)}{\partial \theta_{s,a'}} = -\frac{1}{\tau}\pi_\theta(s,a') \approx -\frac{1}{\tau}\hat{\pi}_\theta(s,a')$. If a state $s$ does not appear in any trajectory in $X$, then $\mathbb{N}(s) = 0$, and there is no information to estimate $\pi(s,a)$ for any $a \in A$. However, given the policy gradient estimate $DJ_1(\theta)$ is computed from the sampled trajectory, then the term $\nabla \log \pi_\theta(s,a)$ will not appear in the gradient estimate for unseen state $s$.

We also need to find the gradient $DQ^\star(R_2(x))$. For this step, we need to estimate $\frac{\partial Q^\star}{\partial r_{\tilde{s},\tilde{a}}}$ for each $(\tilde{s},\tilde{a})$ where $\frac{\partial R_2(x)_{\tilde{s},\tilde{a}}}{\partial x_{s',a'}} \neq 0$ for any $s',a'$. This step of calculating $\frac{\partial Q^\star_{s,a}}{\partial r_{\tilde{s},\tilde{a}}}$ is equivalent to a data-driven off-policy evaluation given the reward function $r(s,a) = 1$ if $(s,a) = (\tilde{s},\tilde{a})$ and $r(s,a) = 0$ otherwise. So, we can use Monte Carlo policy evaluation (Precup, 2000) to construct an unbiased estimator $\overline{DQ^\star(R_2(x))}$ of $DQ^\star(R_2(x))$.

To make sure that the estimated gradient is unbiased, that is,

$$\mathbb{E}\left(\overline{DJ_1(Q^\star(R_2(x)))} \cdot \overline{DQ^\star(R_2(x))}\right) = DJ_1(Q^\star(R_2(x))) \cdot DQ^\star(R_2(x)),$$

we use two different samples $X^1$ and $X^2$ to estimate the value of $\overline{DJ_1(Q^\star(R_2(x)))}$ and $\overline{DQ^\star(R_2(x))}$ respectively. This operation ensures that two estimates are uncorrelated.

---

**Algorithm 2** Adaptive incentive design without knowing P2's reward

---

1: **procedure** ADAPTIVE INCENTIVE DESIGN WITHOUT KNOWING P2'S REWARD( )
2:     Initialize $x_0, \eta, \delta_0 > 0$;           ▷ $\eta$ is the step size, $\delta_0$ is a termination threshold and is close to 0.
3:     $k \leftarrow 0$;
4:     Collect two sets of P2's trajectories $X_0^1, X_0^2$ given P2's best response to $R_2(x_0)$;
5:     $J_1^0 \leftarrow \frac{1}{|X_0^1 \cup X_0^2|} \sum_{\rho \in X_0^1 \cup X_0^2} R_1(\rho) - h(x_0)$;
6:     $\delta \leftarrow \infty$;
7:     **while** $\delta \geq \delta_0$ **do**
8:         Estimate $\overline{DJ(Q^\star(R_2(x_k)))}$ from $X_k^1$;
9:         Estimate $\overline{DQ^\star(R_2(x_k))}$ using Monte Carlo policy evaluation from $X_k^2$;
10:        $\overline{DJ(x_k)} \leftarrow \overline{DJ_1(Q^\star(R_2(x_k)))} \cdot \overline{DQ^\star(R_2(x_k))} \cdot DR_2(x_k) - Dh(x_k)$;
11:        $x_{k+1} \leftarrow \mathsf{Proj}_{\mathbf{R}_+^{|S \times A|}}(x_k + \eta \overline{DJ(x_k)})$                 ▷ Projection to non-negative orthant.
12:        Collect new trajectories $X_{k+1}^1, X_{k+1}^2$ given P2's best response to $R_2(x_{k+1})$;
13:        $X_{k+1} \leftarrow X_{k+1}^1 \cup X_{k+1}^2$;
14:        $J_1^{k+1} \leftarrow \frac{1}{|X_{k+1}|} \sum_{\rho \in X_{k+1}} R_1(\rho) - h(x_{k+1})$;
15:        $\delta \leftarrow |J_1^{k+1} - J_1^k|$;
16:        $k \leftarrow k + 1$;
17:    **end while**
18:    **return** $x_k$
19: **end procedure**

---

Algorithm 2 describes an adaptive incentive design without knowing P2's reward function. At the $k$-th iteration, P1 employs a side payment $x_k$ and observes two sets $X_k^1, X_k^2$ of sampled trajectories from P2's best response given $R_2(x_k)$. The gradient $DJ(x_k)$ of the objective function is estimated from the two sets of samples and then used to compute the updated side payment $x_{k+1}$. The algorithm terminates once P1's value does not change more than $\delta_0$ between two consecutive iterations.

Next, we show that without knowing the follower's reward function, our algorithm still converges to a local optimal solution.

**Proposition 4.** *The gradient estimate $\overline{DJ(x)}$ is unbiased.*

**Theorem 2.** *Let $(x_k)_{k \in \mathbb{N}}$ be a sequence of side payments generated by adaptive incentive design in Algorithm 2. With the step size $\eta = \frac{\sqrt{2}}{L\sqrt{T}}$, we have that*

$$\min_{k=0,\ldots,T-1} \mathbb{E}\|DJ(x_k)\|^2 \leq \frac{2\sqrt{2}L(\sup_{x \in \mathbf{R}_+^{|S \times A|}} J(x) - J(x^0))}{\sqrt{T}}.$$

*Equivalently, for a given $\epsilon > 0$, it requires $T = \mathcal{O}(\epsilon^{-2})$ iterations to ensure $\min_{k=0,\ldots,T-1} \mathbb{E}\|DJ(x_k)\|^2 \leq \epsilon$.*

## 4 Incentive-invariant reward shaping

Our proposed adaptive incentive design requires no knowledge or learning of P2's reward function. An alternative approach is to learn P2's reward function from observed trajectories of P2, using inverse reinforcement learning methods (Abbeel & Ng, 2004; Ng & Russell, 2000; Ramachandran & Amir, 2007; Ziebart et al., 2008), and then apply the incentive design with the learned P2's reward case. For more information about inverse reinforcement learning algorithms, the readers are referred to the survey by Arora & Doshi (2021).

Let's refer the second method as *reward learning-based incentive design*. However, it is known that inverse reinforcement learning is ill-posed because there can be multiple reward functions that generate the same optimal policy (Arora & Doshi, 2021). In that case, no amount of data can distinguish them. These reward functions are known as *policy invariant*.

For the reward-learning-based incentive design, we are interested in the following question: Given two reward functions $R_2$ and $R_2^\dagger$ that are policy-invariant to each other, are these two rewards also *incentive-invariant*? Namely, is the optimal incentive design $x$ for P2 with a reward function $R_2$ also an optimal incentive design for P2 whose reward function is $R_2^\dagger$?

First, we consider a class of policy-invariant reward functions defined by potential-based reward shaping (Skalse et al., 2023; Ng et al., 1999).

**Definition 1** (Potential shaping). *A potential function is a function $\phi : S \to \mathbf{R}$, where $\phi(s) = 0$ if $s$ is a sink/absorbing state. Let $R_2$ and $R_2^\dagger$ be reward functions. The $R_2^\dagger$ is produced by a $\phi$-potential-based reward shaping of $R_2$ if*

$$R_2^\dagger(s,a) = R_2(s,a) + \gamma \mathbb{E}_{s' \sim P(\cdot|s,a)}\phi(s') - \phi(s).$$

For simplicity, let $F(s,a) = \gamma \mathbb{E}_{s' \sim P(\cdot|s,a)}\phi(s') - \phi(s)$ be the shaping reward function.

The following results are shown in Skalse et al. (2023) and paraphrased.

**Theorem 3** (Skalse et al. (2023)). *Given an MDP $M = \langle S, A, P, \mu, \gamma, R \rangle$ and a temperature parameter $\tau$, the entropy-regulated policy $\pi$ determines $R$ up to potential-based reward shaping.*

In other words, if $R_2^\dagger$ is a $\phi$-potential-based reward shaping of $R_2$, then an entropy-regulated optimal policy $\pi$ with respect to $R_2$ is also entropy-regulated optimal policy given the reward function $R_2^\dagger$.

Next, we prove the following theorem.

**Theorem 4.** *Given an MDP $M$ and a temperature parameter $\tau$, the leader's reward function $R_1$ and side payment cost function $h$, if P2 selects an entropy-regulated optimal policy given reward $R_2(x)$ for any side payment $x$, then the optimal incentive design $x^\star$ for P2 with reward $R_2$ is optimal for P2 with reward $R_2^\dagger$.*

The above incentive-invariance condition holds when P2 selects the entropy-regulated optimal policy given the reward function. However, the result naturally generalizes to the case when P2 selects a maximal supportive optimal policy with respect to the reward function. An optimal policy is *maximally supportive* if it takes all optimal actions with positive probability. The maximally supportive optimal policy is not unique.

**Corollary 1.** *Given an MDP $M$ and a temperature parameter $\tau$, the leader's reward function $R_1$ and side payment cost function $h$, if P2 selects a maximally supportive optimal policy given the reward $R_2(x)$ for any incentive design $x$ and always break the tie in the favor for P1 (formal definition in the proof), then the optimal incentive design $x^\star$ for P2 with reward $R_2$ is optimal for P2 with reward $R_2^\dagger$.*

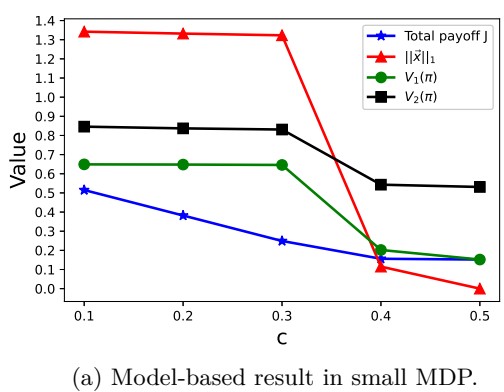

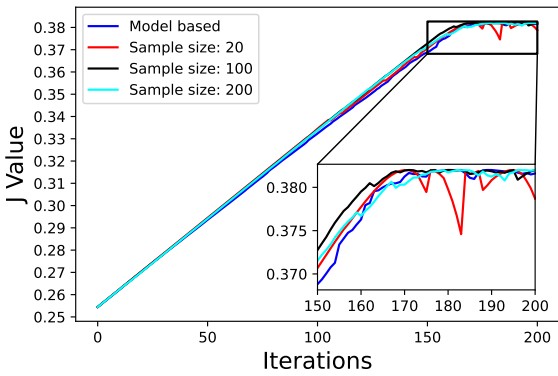

(a) Model-based result in small MDP.

(b) Model-free results in small MDP.

Figure 3: [Left] Incentive design with known follower's reward functions and different cost functions for side payments. [Right] The convergence results of adaptive incentive design with unknown follower's reward functions, with different sample sizes used for gradient estimation. All results are for the small MDP example.

## 5 Experiments

We illustrate the proposed methods with two sets of examples, one is a probabilistic graph-based MDP and another is a stochastic gridworld [1]. For all case studies, the workstation used is powered by Intel i7-11700K and 32GB RAM. In experiments, we constrain the leader to allocate a subset of state-action pairs with side-payments. This constraint is to reduce the computation for gradient calculation (recall the complexity analysis in Section 3.1) and can be removed if more powerful GPUs are used.

### 5.1 An MDP Example

We consider first a small MDP in which the follower has four actions {"a","b","c","d"}. For clarity, the graph in Fig. 2 only shows the transition given action $a$ where a thick (resp. thin) arrow represents a high (resp. low) transition probability, self-loops are omitted. For example, $P(0, "a") = \{1 : 0.7, 2 : 0.1, 3 : 0.1, 4 : 0.1\}$. Once the follower enters states $\{10, 11, 12\}$, the follower enters the "Sink" state with probability 1 by taking any actions. The follower only receives a reward of 1 when he visits state 10. The leader only obtains a reward of 1 when the follower visits state 11. The discounting factor $\gamma$ is 0.95 in this example. In order to incentivize the follower to visit state 11, the leader offers side-payment to the follower if the follower visits state 11 and takes action $a$, which means the domain of variable $x$ is confined to $\mathcal{X} = \{x \mid x(s, a) = 0, \forall (s, a) \neq (11, a), x(11, a) \geq 0\}$. The follower's reward with the side-payment is simply the sum $R_2(s, a, x) = R_2(s, a) + x(s, a)$, for all $(s, a) \in S \times A$.

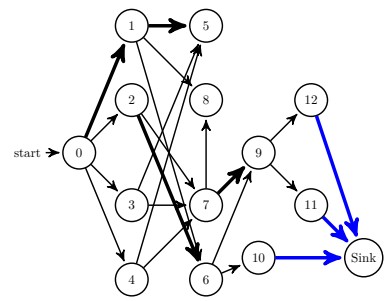

Figure 2: Illustrating the graph of a small MDP.

We consider the cost function of making side-payment in the form of $h(x) = c\|x\|_1$. We conduct the experiment for $c \in \{0.1, 0.2, 0.3, 0.4, 0.5\}$. Figure 3a shows the result of the leader's total payoff($J(\nu, x, \pi)$), the leader's expected value($V_1(\pi)$), the follower's expected value ($V_2(\pi)$) given the follower's best response policy and the reward $R_2(\cdot, x)$, and the value of $\|x\|_1$. When the leader does not offer any side-payment, the leader's expected value is 0.152, and the follower's expected value is 0.531. In this case, the follower visits 11 due to the probabilistic outcomes of the policy.

---

[1]Code: `https://github.com/alexalvis/IncentiveFollower`

It is observed that when $c \in \{0.1, 0.2, 0.3\}$, the leader's side-payment policies $h(x)$ are similar, where $x(11, \text{"a"}) \approx 1.33$. These similar side-payment policies result in similar best response policies from the follower, and thus the similar value $V_1(\pi) \approx 0.648$ for the leader. However, the total payoff of the leader decreases due to the increasing cost of side-payment. As $c$ increases to 0.4, the leader's side-payment policy is $x(11, \text{"a"}) = 0.108$, and the corresponding leader's value $V_1(\pi)$ is 0.198, and the total payoff $J$ is 0.155.

If the leader's side-payment policy given $c = 0.4$ is $x(11, \text{"a"}) \approx 1.33$ (used when $c = 0.3$), the total payoff $J$ is 0.127, which indicates the cost of side-payment outweighs the benefit to the total payoff. Finally, when $c$ increases to 0.5, the leader does not offer any incentive. As $c$ ranges from 0.1 to 0.5, the follower's value $V_2(\pi)$ given the best response policy $\pi$ to reward $R_2(\cdot, x)$ decreases due to the decrease in the side-payment. When $c = 0.4$, the follower's value is 0.543. When $c = 0.5$, the leader stops offering any side-payment and the follower's expected value is the lowest 0.531, recovering the original follower's value without any incentives.

Next, we consider the scenario, when the follower's reward is unknown to the leader. We fix $c$ to be 0.2. Figure 3b shows the value of leader's total payoff converges over iterations, given different trajectory sample sizes used in the gradient estimates. Our result shows that in the first 150 iterations, the convergence trend is stable in all three cases, compared to the model-based approach. After 150 iterations, oscillations are observed with a small sample size (20). After 200 iterations, the algorithm converges for all three cases with different sample sizes. The result indicates that the leader can obtain the optimal incentive design without estimating the follower's true reward function.

## 5.2 Gridworld Example

In the second example, we consider a robot motion planning problem in a stochastic gridworld. Consider a six by six gridworld in Figure 4. The robot aims to reach a set of goal states while maximizing its payoff. The robot can move in four compass directions. Given an action, say, "N", the robot enters the intended cell with $1 - 2\alpha$ probability and enters the neighboring cells, which are west and east cells with $\alpha$ probability. In our experiments, $\alpha$ is selected to be 0.1. Each state is identified by its coordinates $(i, j)$, denoting the state at row $i$ and column $j$. The robot's task is to reach one of the green or blue cells. Entering a green cell yields a reward of 10, while entering a blue cell results in a reward of 8. The process terminates once the robot enters one of these blue or green cells. Upon entering the states with fire, the robot incurs a punishment of $-5$. The leader only receives a reward of 10 when the robot enters a blue cell. Thus, the leader is to incentivize the robot to visit the blue cells. The discounting factor $\gamma$ is 0.95.

We first conduct the experiment on the model-based case. The cost function of making side-payment is in the form of $h(x) = c\|x\|_1$. Assume the leader can incentivize the robot with side-payment $x$, constrained by $x \in \mathcal{X} = \{x \mid x(s, a) = 0, \forall (s, a) \neq ((0, 3), \text{"N"})$ and $x((0, 3), \text{"N"}) \geq 0\}$. The experiment results under various $c$ values are shown in Figure 5a. When $c$ is less than 1.1, the side-payment value and the leader's value slowly decrease with the increase of the $c$. However, when $c$ reaches 1.1, the leader opts for a strategy of assigning 0 side payment to the state-action pair $((0, 3), \text{"N"})$. If the leader were to maintain the same side-payment policy as the policy when $c$ equals 1.0, the resulting total payoff $J$ is negative. That indicates placing side-payment at state $((0, 3), \text{"N"})$ becomes disadvantageous for the leader's overall payoff when $c = 1.1$.

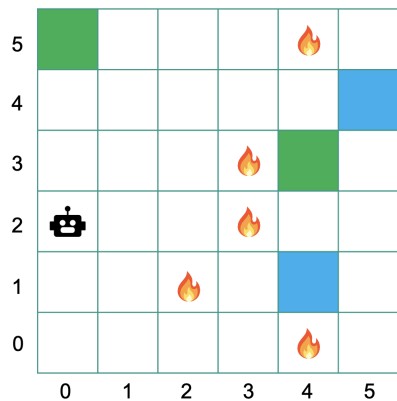

Figure 4: A $6 \times 6$ gridworld.

We also conduct the experiment under a different constraint set $\mathcal{X}' = \{x \mid x(s, a) = 0, \forall s \notin \{(1, 4), (4, 5)\}, x(s, a) \geq 0, \text{ for } (s, a) \in \{(1, 4), (4, 5)\} \times A\}$. Our intuition is that the robot selects to reach a cell where it obtains the highest reward. Thus assigning side-payment to other states only incurs a cost to the leader's total payoff $J$. Given the environment, it costs less side-payment to attract the robot to $(1, 4)$, compared to $(4, 5)$. As shown in Figure 5b, for $0.1 \leq c \leq 0.7$, the leader achieves a higher expected value compared to the result when the leader allocates side-payment to $((0, 3), \text{"N"})$. But the leader also

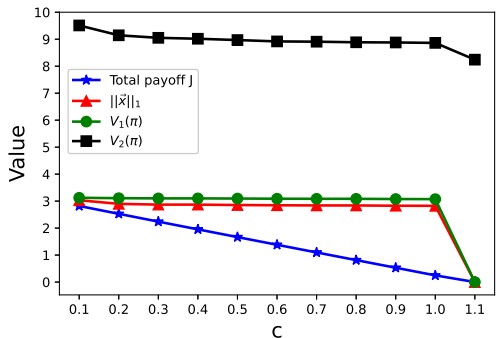

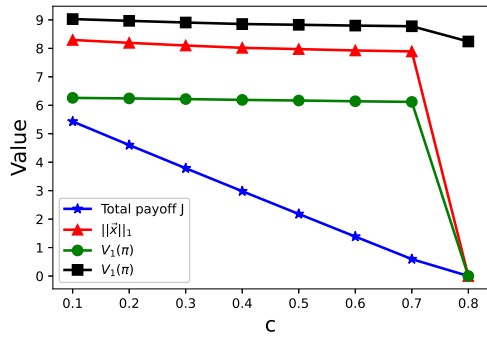

(a) Model-based result in gridworld, given the leader assigns side payment at $((0,3), \text{"N"})$.

(b) Model-based result in gridworld, given the leader assigns side-payment at $\{(1,4),(4,5)\} \times A$.

Figure 5: Incentive design results with known follower's reward functions in the grid world, with different side payment cost functions.

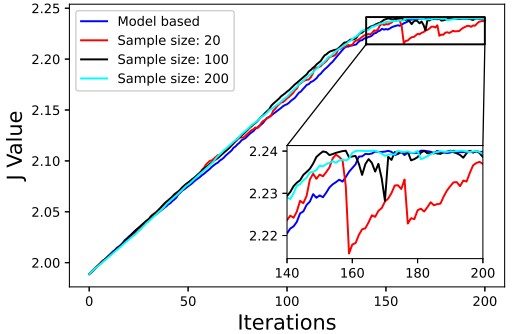

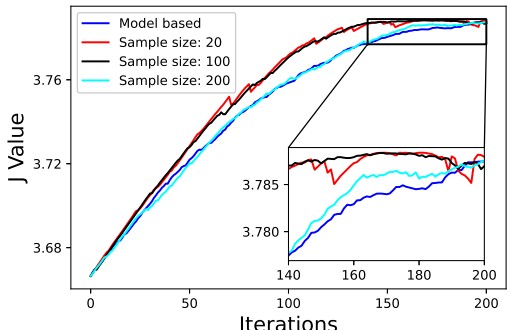

(a) Model-free solution convergence trend in gridworld, given the leader assigns side payment at $((0,3), \text{"N"})$.

(b) Model-free solution convergence trend in gridworld, given the leader assigns side-payment at $\{(1,4),(4,5)\} \times A$.

Figure 6: The convergence results of adaptive incentive design with unknown follower's reward functions in the gridworld, with different sample sizes used for gradient estimation.

allocates more resources, so a higher leader's expected value $V_1(\pi)$ does not necessarily mean a higher value for the total payoff $J$. In particular, when $c$ increases to 0.8, the leader assigns 0 side payment, resulting in the leader's value approaching 0. But in the previous case with the constraint set $\mathcal{X}$, the leader can still allocate side-payment to incentivize the follower, which leads to a total payoff $J = 0.814$ when $c = 0.8$. These two sets of experiments indicate the significance of the state-action pairs chosen by the leader for allocating side-payments.

Next, we look into the model-free scenarios. Let us fix $c$ to be 0.3 and assume the leader allocates side-payments to $((0,3), \text{"N"})$. We conduct the experiments with different trajectory sample sizes and illustrate the results in Figure 6a. As depicted, the results of the gradient-ascent algorithm for different sample sizes are similar in the initial 100 iterations. However, after 100 iterations, noticeable oscillations are observed, particularly when the sample size is 20. The oscillations also exist when the sample size is 100 but not that significant. When the sample size is 200, the convergence result over iterations is close to the model-based scenario. The final converged side-payment policies are similar with model-based methods across all cases.

Similar experiments are conducted given $c = 0.3$ and the leader allocates side-payment to $\{(1,4),(4,5)\} \times A$. We show the results in Figure 6b and the results of the value versus iterations are similar to the previous

cases. Due to the property of stochastic gradient ascent, a larger sample size corresponds to a more stable convergence.

## 6 Conclusion

We develop a gradient-based method for designing an optimal incentive policy that aims to motivate the follower to choose policies in favor of the leader. The gradient-based incentive design hinges upon computing the gradient of the follower's policy parameters with respect to the leader's incentive decision variables. We show that if the follower uses an entropy-regulated optimal policy, then the exact gradient can be computed in polynomial time. In addition, the gradient can be estimated from the sampled trajectories generated by the follower's response. As a result, our method naturally provides an adaptive incentive design method where the leader can iteratively update his incentive policy upon observing sampled runs from the follower's policy, without the need to infer the follower's reward function. At last, we also show that the optimal incentive policies can be the same for two different followers' reward functions, provided that the two reward functions are related through a policy-invariant reward shaping. We experimentally validate the proposed method and discuss the stability of the convergence given various sample sizes used in gradient estimation.

The limitations of the method would be that if the leader has the flexibility to provide incentive for any state-action pair, then it is computationally expensive to solve the optimal incentive policy. To improve efficiency, we could restrict the size of leader's incentive decision variables by selecting a subset of state-action pairs for allocating side-payments. Experiment results show that the leader's value varies given different sets of selected state-action pairs. How to optimally determine the subsets of state-action pairs for efficient computation is an important topic. Another interesting direction is to robustifying the incentive design with respect to uncertainty in the follower's parameters, for example, the bounded rationality temperature parameter $\tau$ can be unknown and belong to a given range. It is an open question if the incentive design can be robust to small variations in these parameters.

## Acknowledgements

Research was sponsored by NSF under award numbers 2207759 and in part by the Army Research Laboratory under Cooperative Agreement Number W911NF-22-1-0166. The views and conclusions contained in this document are those of the authors and should not be interpreted as representing the official policies, either expressed or implied, of the Army Research Laboratory or the U.S. Government. The U.S. Government is authorized to reproduce and distribute reprints for Government purposes notwithstanding any copyright notation herein.

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

## A  Derivations and Proofs

### A.1  Derivation of policy gradient

The detailed derivation of policy gradient in equation 9 in Proposition 1.

Recall that $\frac{\partial \log \pi_\theta(s,a)}{\partial \theta_{\tilde{s},\tilde{a}}} = 0$ if $s \neq \tilde{s}$ and

$$
\begin{aligned}
\frac{\partial \log \pi_\theta(s,a)}{\partial \theta_{s,a}} &= \frac{\partial \log \exp(\frac{\theta_{s,a}}{\tau})}{\partial \theta_{s,a}} - \frac{\partial \log \sum_{a' \in A} \exp(\frac{\theta_{s,a'}}{\tau})}{\partial \theta_{s,a}} \\
&= \frac{1}{\tau} - \frac{1}{\sum_{a' \in A} \exp(\frac{\theta_{s,a'}}{\tau})} \frac{\partial \exp(\frac{\theta_{s,a}}{\tau})}{\partial \theta_{s,a}} \\
&= \frac{1}{\tau}(1 - \pi_\theta(s,a))
\end{aligned}
\tag{14}
$$

and

$$\frac{\partial \log \pi_\theta(s,a)}{\partial \theta_{s,a'}} = \frac{1}{\tau}(0 - \pi_\theta(s,a')) \tag{15}$$

## A.2  Proof of Proposition 3

To show that $J(x)$ is smooth (Proposition 3), we first prove the following properties:

**Proposition 5.** *Consider $V(\cdot, \theta) : S \to \mathbf{R}$ the value function of a softmax policy with parameter $\theta$. For any $s \in S$, the value function $V(s, \cdot)$ is Lipschitz continuous, which is*

$$\|V(s,\theta_1) - V(s,\theta_2)\| \le L_c\|\theta_1 - \theta_2\|.$$

*where $L_c$ is the Lipschitz constant.*

*Proof.* Using the policy gradient with softmax policy, $\frac{\partial V(s,\theta)}{\partial \theta_{s,a}} = \frac{1}{1-\gamma}d_\mu^{\pi_\theta}(s)\pi_\theta(s,a)A^{\pi_\theta}(s,a)$, where $d_\mu^{\pi_\theta}(s) = (1-\gamma)\sum_{t=0}^{\infty}\gamma^t P^{\pi_\theta}(s_t = s|s_0 \sim \mu)$ is the discounted state visitation distribution. Thus,

$$\|\nabla_\theta V(s,\theta)\| = \|\frac{1}{1-\gamma}d_\mu^{\pi_\theta}(s)\pi_\theta(s,a)A^{\pi_\theta}(s,a)\| \tag{16}$$

$$= \frac{1}{1-\gamma}\sqrt{\sum_{s,a}(d_\mu^{\pi_\theta}(s)\pi_\theta(s,a)A^{\pi_\theta}(s,a))^2} \tag{17}$$

$$\le \frac{|\max_{s,a} A^{\pi_\theta}(s,a)|}{1-\gamma}\|(d_\mu^{\pi_\theta}(s)\pi_\theta(s,a))\|_2 \tag{18}$$

$$\le \frac{|\max_{s,a} A^{\pi_\theta}(s,a)|}{1-\gamma}\|(d_\mu^{\pi_\theta}(s)\pi_\theta(s,a))\|_1 \tag{19}$$

$$\le \frac{|R_{max}|}{1-\gamma}, \tag{20}$$

where the last step is because $\|(d_\mu^{\pi_\theta}(s)\pi_\theta(s,a))\|_1 = \|d_\mu^{\pi_\theta}(s)\|_1 \le 1$. $|R_{max}|$ is the maximum possible value of the reward function.

Let $L_c = \frac{|R_{max}|}{1-\gamma}$, we have

$$\|V(s,\theta_1) - V(s,\theta_2)\| \le L_c\|\theta_1 - \theta_2\|, \quad \forall s \in S.$$

$\square$

**Proposition 6.** *The function $J_1(\theta) \triangleq V_1(\mu, \theta)$ is $L_\theta$-smooth, that is,*

$$\|\nabla_\theta J(\theta) - \nabla_\theta J(\theta')\| \le L_\theta\|\theta - \theta'\|_2,$$

*where $L_\theta$ is a Lipschitz constant.*

The proof of Proposition 6 is based on Lemma 55 in Agarwal et al. (2021) by setting the coefficient before the barrier function to 0. Thus, we omit the detailed proof here.

**Proposition 7.** *Let $Q^\star(R_2(x))$ be the entropy-regulated optimal state-action value function in the MDP $M(x)$, it holds that $Q^\star(R_2(x))$ is Lipschitz continuous in $x$, that is,*

$$\|Q^\star(R_2(x_1)) - Q^\star(R_2(x_2))\| \le L_x\|x_1 - x_2\|.$$

*where $L_x$ is the Lipschitz constant.*

*Proof.* For notational convenience and clarity, let $Q^\star(x) \triangleq Q^\star(R_2(x))$.

First, we start from the infinity norm term $\|Q^\star(x_1) - Q^\star(x_2)\|_\infty$, it is noted that $\|Q^\star(x_1) - Q^\star(x_2)\|_\infty = \max_{s,a}|Q^\star(s,a,x_1) - Q^\star(s,a,x_2)|$.

By the definition of entropy-regularized state-action value function,

$$Q^\star(s,a,x) = \mathbb{E}_{\pi_{Q^\star(x)}}\left[\sum_{t=0}^\infty \gamma^t R_2(S_t, A_t; x) - \tau \log \pi_{Q^\star(x)}(S_t, A_t) \,\Big|\, S_0 = s, A_0 = a\right],$$

Thus,

$$
\begin{aligned}
&Q^\star(s,a,x_1) - Q^\star(s,a,x_2)\\
=&\mathbb{E}_{\pi_{Q^\star(x_1)}}\left[\sum_{t=0}^\infty \gamma^t R_2(S_t, A_t; x_1) - \tau \log \pi_{Q^\star(x_1)}(S_t, A_t) \,\Big|\, S_0 = s, A_0 = a\right]\\
&- \mathbb{E}_{\pi_{Q^\star(x_2)}}\left[\sum_{t=0}^\infty \gamma^t R_2(S_t, A_t; x_2) - \tau \log \pi_{Q^\star(x_2)}(S_t, A_t) \,\Big|\, S_0 = s, A_0 = a\right]\\
=&\mathbb{E}_{\pi_{Q^\star(x_1)}}\left[\sum_{t=0}^\infty \gamma^t R_2(S_t, A_t; x_1) - \tau \log \pi_{Q^\star(x_1)}(S_t, A_t) \,\Big|\, S_0 = s, A_0 = a\right]\\
&- \mathbb{E}_{\pi_{Q^\star(x_1)}}\left[\sum_{t=0}^\infty \gamma^t R_2(S_t, A_t; x_2) - \tau \log \pi_{Q^\star(x_1)}(S_t, A_t) \,\Big|\, S_0 = s, A_0 = a\right]\\
&+ \mathbb{E}_{\pi_{Q^\star(x_1)}}\left[\sum_{t=0}^\infty \gamma^t R_2(S_t, A_t; x_2) - \tau \log \pi_{Q^\star(x_1)}(S_t, A_t) \,\Big|\, S_0 = s, A_0 = a\right]\\
&- \mathbb{E}_{\pi_{Q^\star(x_2)}}\left[\sum_{t=0}^\infty \gamma^t R_2(S_t, A_t; x_2) - \tau \log \pi_{Q^\star(x_2)}(S_t, A_t) \,\Big|\, S_0 = s, A_0 = a\right]\\
\overset{(i)}{\leq}\;&\mathbb{E}_{\pi_{Q^\star(x_1)}}\left[\sum_{t=0}^\infty \gamma^t R_2(S_t, A_t; x_1) - \tau \log \pi_{Q^\star(x_1)}(S_t, A_t) \,\Big|\, S_0 = s, A_0 = a\right]\\
&- \mathbb{E}_{\pi_{Q^\star(x_1)}}\left[\sum_{t=0}^\infty \gamma^t R_2(S_t, A_t; x_2) - \tau \log \pi_{Q^\star(x_1)}(S_t, A_t) \,\Big|\, S_0 = s, A_0 = a\right]\\
=&\mathbb{E}_{\pi_{Q^\star(x_1)}}\left[\sum_{t=0}^\infty \gamma^t \left(R_2(S_t, A_t; x_1) - R_2(S_t, A_t; x_2)\right) \,\Big|\, S_0 = s, A_0 = a\right]\\
\leq\;&\mathbb{E}_{\pi_{Q^\star(x_1)}}\left(\sum_{t=0}^\infty \gamma^t \max_{s',a'}|x_1(s',a') - x_2(s',a')|\right)\\
=&\frac{1}{1-\gamma}\|x_1 - x_2\|_\infty,
\end{aligned}
$$

where inequality $(i)$ is because $\pi_{Q^\star(x_2)}$ is optimal w.r.t. the reward function $R_2(x_2)$.

Due to the equivalence of norms Conrad (2018), on the finite dimension C, for all $x \in C$, $B\|x\|_\infty \leq \|x\|_1 \leq B'\|x\|_\infty$ where $B, B'$ are two constants. We can rewrite the infinity norm case we used above to 1-norm case. Thus, we have $\|Q^\star(R_2(x_1)) - Q^\star(R_2(x_2))\| \leq L_x\|x_1 - x_2\|$. $\qquad\square$

**Remark 2.** *Proposition 7 holds for a more general case when for any state-action pair $(s,a) \in S \times A$, the reward function $R_2(s,a,x) = R_2(s,a) + g(s,a,x)$ for any Lipschitz continuous function $g(\cdot)$.*

**Proposition 8.** *$DQ^\star(R_2(x)) \cdot DR_2(x)$ is Lipschitz continuous with respect to $x$, which means*

$$\|DQ^\star(R_2(x_1)) \cdot DR_2(x_1) - DQ^\star(R_2(x_2)) \cdot DR_2(x_2)\| \leq L_\beta\|x_1 - x_2\|.$$

*where $L_\beta$ is the Lipschitz constant.*

*Proof.* By Proposition 2, it is noted that

$$\frac{\partial Q^\star(s,a,r)}{\partial r(\tilde{s},\tilde{a})} = \mathbb{E}_{\pi_{Q^\star(r)}}\left(\sum_{t=0}^\infty \gamma^t \hat{r}_{\tilde{s},\tilde{a}}(S_t, A_t) \,\Big|\, S_0 = s, A_0 = a\right) = Q^{\hat{r}_{\tilde{s},\tilde{a}}}(s,a,\pi_{Q^\star(r)}),$$

where $Q^{\hat{r}_{\tilde{s},\tilde{a}}}(\cdot \pi_{Q^{\star}(r)})$ is the state-action value function given the policy $\pi_{Q^{\star}(r)}$ and reward function $\hat{r}_{\tilde{s},\tilde{a}}$ defined by $\hat{r}_{\tilde{s},\tilde{a}}(s,a) = \mathbf{1}_{\tilde{s},\tilde{a}}(s,a)$. Thus

$$\|DQ^{\star}(R_2(x_1)) - DQ^{\star}(R_2(x_2))\| = \|Q^{\hat{r}_{\tilde{s},\tilde{a}}}(\pi_{Q^{\star}(R_2(x_1))}) - Q^{\hat{r}_{\tilde{s},\tilde{a}}}(\pi_{Q^{\star}(R_2(x_2))})\| \tag{21}$$
$$\leq L_c \|Q^{\star}(R_2(x_1)) - Q^{\star}(R_2(x_2))\|. \tag{22}$$

Equation 22 is derived based on Proposition 5 where the Lipschitz constant $L_c = \frac{1}{1-\gamma}$. This is because the reward function is an indicator function and $R_{max} = 1$ and the entropy-regularized optimal policy $\pi_{Q^{\star}(R_2(x))}$ has a softmax parameterization with a policy parameter $Q^{\star}(R_2(x))$.

Combining equation 22 with Proposition 7, we have $\|DQ^{\star}(R_2(x_1)) - DQ^{\star}(R_2(x_2))\| \leq L_c L_x \|x_1 - x_2\|$.

Since $R_2(x)$ is a linear combination of the original follower's reward function $R_2$ and a Lipschitz continuous function $f$ of $x$. Let $f$ be Lipschitz continuous with a Lipschitz constant $L_f$, then we have $DR_2(x) \leq L_f$. In a special case when $R_2(x) = R_2 + L_f \cdot x$, then $D\boldsymbol{r}(x) = L_f$. Thus, $\|DQ^{\star}(R_2(x_1)) \cdot DR_2(x_1) - DQ^{\star}(R_2(x_2)) \cdot DR_2(x_2)\| \leq L_f L_c L_x \|x_1 - x_2\|$. The proof is completed by letting $L_\beta = L_f L_c L_x$. $\qquad \square$

In the next, we show a general result and the Lipschitz continuity of the term $DJ_1(Q^{\star}(R_2(x))) \cdot DQ^{\star}(R_2(x)) \cdot DR_2(x)$ can be derived by letting $f(x) = DJ_1(Q^{\star}(R_2(x)))$ and $g(x) = DQ^{\star}(R_2(x)) \cdot DR_2(x)$.

**Lemma 2.** *Let $f \colon \mathcal{X} \to \mathbf{R}^{m \times n}$ and $g \colon \mathcal{X} \to \mathbf{R}^n$ be two Lipschitz continuous functions with $\|f(x)\| \leq \overline{f}$ and $\|g(x)\| \leq \overline{g}$ for all $x \in \mathcal{X}$. Then $f \cdot g$ is Lipschitz continuous.*

*Proof.* Since $f$ and $g$ are Lipschitz continuous, there exist $L_f$ and $L_g$ such that that $\|f(x) - f(x')\| \leq L_f \|x - x'\|$ and $\|g(x) - g(x')\| \leq L_g \|x - x'\|$ for all $x, x' \in \mathcal{X}$. For any $x, x' \in \mathcal{X}$, one can obtain

$$\begin{aligned}
\|f(x)g(x) &- f(x')g(x')\| \\
&= \|f(x)g(x) - f(x')g(x) + f(x')g(x) - f(x')g(x')\| \\
&\leq \|f(x)g(x) - f(x')g(x)\| + \|f(x')g(x) - f(x')g(x')\| \\
&\leq \|f(x) - f(x')\|\|g(x)\| + \|f(x')\|\|g(x) - g(x')\| \\
&\leq \overline{g} L_f \|x - x'\| + \overline{f} L_g \|x - x'\| \\
&= (\overline{g} L_f + \overline{f} L_g)\|x - x'\|,
\end{aligned}$$

which implies that $f \cdot g$ is Lipschitz continuous.

$\qquad \square$

Finally, we are ready to prove Proposition 3.

*Proof.* Function $J(x)$ is non-concave but differentiable in our setting. Its total derivative $DJ(x)$ is Lipschitz continuous because

1) $Dh(x)$ is Lipschitz-continuous due to the assumption of $h(x)$ (see Problem 1).

2) $DJ_1(Q^{\star}(R_2(x)))$ is a Lipschitz continuous function and upper bounded by $\frac{R_{max}}{1-\gamma}$ (Proposition 5);

3) $DQ^{\star}(R_2(x)) \cdot DR_2(x)$ is a Lipschitz continuous function (Proposition 8). Furthermore, because $\|DQ^{\star}(R_2(x))\|_\infty$ is the norm of state-action value vector given policy $\pi_{Q^{\star}(R_2(x))}$ and indicator reward functions, it is upper bounded by 1. $\|DR_2(x)\|_\infty \leq L_f$ given $R_2(s,a,x) = \bar{R}_2 + f(s,a,x)$ where $f$ is $L_f$-Lipschitz continuous. Thus, $\|DQ^{\star}(R_2(x)) \cdot DR_2(x)\|$ is upper bounded.

4) $DJ_1(Q^{\star}(R_2(x))) \cdot DQ^{\star}(R_2(x)) \cdot DR_2(x)$ is Lipschitz continuous due to the aforementioned properties 2) and 3) and Lemma 2, where $f(x)$ is $DJ_1(Q^{\star}(R_2(x)))$ and $g(x)$ is $DQ^{\star}(R_2(x)) \cdot DR_2(x)$.

$\qquad \square$

## A.3 Proof of Theorem 1

*Proof.* Given the update rule we used in our algorithm $x_{k+1} = \mathsf{Proj}_{\mathbf{R}_+^{|S \times A|}}(x_k + \eta \nabla J(x))$. We want to show as $k \to \infty$, $x_k$ converges to a critical point. Following the Taylor expansion of function $J$ around $x_k$, for a sufficient small step size $\eta$, we have

$$J(x_{k+1}) - J(x_k) \geq \eta \|\nabla J(x_k)\|^2 - \frac{L\eta^2}{2} \|\nabla J(x_k)\|^2 \tag{23}$$

$$= (1 - \frac{L\eta}{2})\eta \|\nabla J(x_k)\|^2. \tag{24}$$

Using $\eta \leq \frac{1}{L}$, we have $(1 - \frac{L\eta}{2}) \geq \frac{1}{2}$. Thus we have $J(x_{k+1}) - J(x_k) \geq \frac{\eta}{2} \|\nabla J(x_k)\|^2$. Which means $J(x_{k+1}) - J(x_k)$ is always positive. Moreover, function $J(\cdot)$ is bounded due to the value function of the leader $V_1^\mu(\theta)$ is bounded. Therefore the function $J(x)$ is monotonically increasing and bounded. Due to the Bolzano-Weierstrass Theorem, every bounded sequence of real numbers has a convergent subsequence thus, the gradient-based method converges to a local optimal point of the function $J(x)$. $\qquad\square$

## A.4 Proof of Theorem 2

*Proof.* Consider the estimated gradient from the observed P2's trajectories:

$$\overline{DJ(x)} = \overline{DJ_1(Q^\star(R_2(x_t)))} \cdot \overline{DQ^\star(R_2(x_t))} \cdot DR_2(x_t) - Dh(x_t)$$

First, the term $\overline{DJ_1(Q^\star(R_2(x_t)))}$ is the estimated policy gradient with respect to P1's reward function. For notational convenience, let $\theta \triangleq Q^\star(R_2(x_t))$ be the entropy-regularized state-action value function.

Since P2's reward function is unknown, we replace the exact gradient term $\nabla \log \pi_\theta(\rho)$ in equation 8 with the approximated gradient $\nabla \log \hat{\pi}_\theta(\rho)$. The bias of the estimator is

$$\mathbb{E}\left( DJ_1(\theta) - \overline{DJ_1(\theta)} \right) \tag{25}$$

$$= \mathbb{E}\left( \int \pi_\theta(\rho) R_1(\rho) \nabla \log \pi_\theta(\rho) d\rho - \int \pi_\theta(\rho) R_1(\rho) \nabla \log \hat{\pi}_\theta(\rho) d\rho \right)$$

$$= \mathbb{E}\left( \int \pi_\theta(\rho) R_1(\rho) \left( \nabla \log \pi_\theta(\rho) - \nabla \log \hat{\pi}_\theta(\rho) \right) d\rho \right)$$

$$= \mathbb{E} \int \pi_\theta(\rho) R_1(\rho) \left( \sum_{i=0}^{|\rho|} \left( \nabla \log \pi_\theta(s_i, a_i) - \nabla \log \hat{\pi}(s_i, a_i) \right) \right) d\rho$$

$$= \int \pi_\theta(\rho) R_1(\rho) \left( \sum_{i=0}^{|\rho|} \mathbb{E} \left( \nabla \log \pi_\theta(s_i, a_i) - \nabla \log \hat{\pi}(s_i, a_i) \right) \right) d\rho$$

$$= 0$$

The last step is due to the $\hat{\pi}(s, a)$ is the unbiased estimator of $\pi(s, a)$. To see why this is the case, consider those non-zero elements in $\nabla \log \pi_\theta(s_i, a_i)$, from equation 9, we have

$$\mathbb{E}\left( \frac{\partial \log \pi_\theta(s, a)}{\partial \theta_{s,a}} - \frac{\partial \log \hat{\pi}_\theta(s, a)}{\partial \theta_{s,a}} \right) = \mathbb{E}\left( \hat{\pi}_\theta(s, a) - \pi_\theta(s, a) \right) = 0.$$

So, $\mathbb{E}\left( \nabla \log \pi_\theta(s_i, a_i) - \nabla \log \hat{\pi}(s_i, a_i) \right) = 0$, and we have our last step equals to 0.

For the second term $\overline{DQ^\star(R_2(x_t))}$, based on Proposition 8, $DQ^\star(R_2(x_t))$ is the policy evaluation of the indicator reward function $\hat{r}_{s,a}(s', a') = \mathbf{1}_{(s,a)}(s', a')$ with respect to the policy $\pi_{Q^\star}$. We use a Monte Carlo policy evaluation for computing $\overline{DQ^\star(R_2(x_t))}$ from sampled trajectory (Precup, 2000), because it is known to be an unbiased estimator of $DQ^\star(R_2(x_t))$.

Next, consider the product term $\overline{DJ_1(Q^\star(R_2(x_t)))} \cdot \overline{DQ^\star(R_2(x_t))}$. Given that both $\overline{DJ_1(Q^\star(R_2(x_t)))}$ and $\overline{DQ^\star(R_2(x_t))}$ are unbiased estimates for $DJ_1(Q^\star(R_2(x_t)))$ and $DQ^\star(R_2(x_t))$ based on our aforementioned analysis. We select two independent samples to estimate $\overline{DJ_1(Q^\star(R_2(x_t)))}$ and $\overline{DQ^\star(R_2(x_t))}$. By doing so, the covariance between these two estimates to be 0. Hence, we arrive at the conclusion that $\overline{DJ(x)}$ is a unbiased estimator of $DJ(x)$. $\qquad\square$

## A.5 Proof of Theorem 4

*Proof.* Given P2's reward $R_2$, denote by $\mathsf{BR}(R_2)$ the entropy-regularized optimal policy (best response) of P2, which is unique. Suppose $x^\star$ is an optimal incentive when P2's reward is $R_2$. This implies $\max_{\pi=\mathsf{BR}(R_2(x^\star))} J(x^\star, \pi) \geq \max_{\pi=\mathsf{BR}(R_2(x))} J(x, \pi)$ for all $x$.

Slightly abusing notation, we refer to $R_2, F, R_2^\dagger$ by their vector representations. Since $F$ is a potential-based reward shaping for $R_2$, by definition it is also a potential-based reward shaping for $R_2(x)$, for any $x$. Due to Theorem 3, we have $\mathsf{BR}(R_2(x)) = \mathsf{BR}(R_2(x) + F) = \mathsf{BR}(R_2^\dagger(x))$ for any $x$ (including $x^\star$). Thus, it follows that $\max_{\pi=\mathsf{BR}(R_2^\dagger(x^\star))} J(x^\star, \pi) \geq \max_{\pi=\mathsf{BR}(R_2^\dagger(x))} J(x, \pi)$ for all $x$, which implies that $x^\star$ is also an optimal incentive design when P2's reward is $R_2^\dagger = R_2 + F$.

$\qquad\square$

## A.6 Proof of Corollary 1

First, the following result is proven in Skalse et al. (2023).

**Theorem 5.** *(Skalse et al., 2023) Given an MDP $M = \langle S, A, P, \mu, \gamma, R \rangle$ and a temperature parameter $\tau$, the maximally supportive optimal policy $\pi$ determines $R$ up to potential-based reward shaping.*

*Proof.* The proof is similar to that of Theorem 4. With a few changes to handle the non-uniqueness in P2's maximally supportive optimal policy. Given P2's reward $R_2$, denote by $\mathsf{BR}_{MS}(R_2)$ the set of maximally supportive optimal policies (best responses) of P2. Suppose $x^\star$ is an optimal incentive when P2's reward is $R_2$. This implies that when P2 breaks the tie in favor of P1, that is, P2 selects $\pi \in \mathsf{BR}_{MS}(R_2(x^\star))$ that maximizes P1's value $J(x^\star, \pi)$, then $\max_{\pi \in \mathsf{BR}_{MS}(R_2(x^\star))} J(x^\star, \pi) \geq \max_{\pi \in \mathsf{BR}_{MS}(R_2(x))} J(x, \pi)$ for all $x$.

Since $F$ is a potential-based reward shaping for $R_2$, by definition it is also a potential-based reward shaping for $R_2 + x$, for any $x$. Due to Theorem 3, we have $\mathsf{BR}_{MS}(R_2(x)) = \mathsf{BR}_{MS}(R_2(x) + F) = \mathsf{BR}_{MS}(R_2^\dagger(x))$ for any $x$ (including $x^\star$). Thus, it follows that $\max_{\pi \in \mathsf{BR}_{MS}(R_2^\dagger(x^\star))} J(x^\star, \pi) \geq \max_{\pi \in \mathsf{BR}_{MS}(R_2^\dagger(x))} J(x, \pi)$ for all $x$, which implies that $x^\star$ is also an optimal incentive design when P2's reward is $R_2^\dagger$.

$\qquad\square$

