# OpenReview forum: "Adaptive Incentive Design for Markov Decision Processes with Unknown Rewards"
_TMLR — Accepted by TMLR_

### Review · Reviewer_DgTw · 2024-10-25

**Summary Of Contributions:**

This paper proposes a gradient-ascent algorithm to compute the leader’s optimal incentive design, without knowing the follower’s reward function. Based on the assumption that the follower’s response satisfies softmax temporal consistency, the paper designs a method to compute the exact gradient of the follower’s policy with respect to the incentive design variables, getting an unbiased estimator of the gradient terms.

**Audience:**

Yes

**Claims And Evidence:**

Yes

**Requested Changes:**

1. Add comparison to other methods in the experiments part. It can also be compared with the ideal case where the follower’s reward function is known, to see the gap between the best possible result.
2. The structure and the notation in the paper can be more clear. What is the relationship between Section 4 Incentive-invariant reward shaping and the main results? Is section 4 proposing another way to estimate the follower’s reward function compared to the Monte Carlo policy evaluation proposed in section 3?

**Strengths And Weaknesses:**

Strengths:
The authors clarify the situation of their work in estimating the leader’s optimal incentive design, without knowing the follower’s reward function, even though the extent of the effectiveness is not certain due to the limit scope of the experiments.

Weakness
1. The bilevel optimization problem is known to be hard to solve and I think it simplifies the problem too much when reduced to the single-level optimization problem, deviating from the original optimization problem.
2. The experiments part is not solid enough to verify the effectiveness of this method, without comparison of other state-of-art methods.

---

> ### Author Response · Authors · 2024-11-06
> **Response to review DgTw**
>
> [Thank you for the suggestions and comments. Please see our response below.]
>
> Weakness
> 1. "The bilevel optimization problem is known to be hard to solve and I think it simplifies the problem too much when reduced to the single-level optimization problem..."
>
> Reply: The proposed method is based on Hypergradient Descent, which is one of the traditional methods for solving bi-level optimization problems. Other bi-level optimization methods include formulating a constrained optimization problem, such as stationary seeking methods-based solutions (see [1][2] for a good discussion of different methods). In this way, all existing bi-level optimization methods reduce the bi-level problem to a single-level or constrained optimization problem, but this reduction does not mean that the problem or the solutions are simplified. Despite the reduction, solving the resulting single-level constrained optimization problem is nontrivial.
>
> As we have shown, when the follower employs an entropy-regularized optimal policy, the lower-level problem admits a unique optimal solution, which allows the use of hypergradient descent. However, hypergradient descent requires computing the derivative of the lower-level optimal solution (policy Q-function) with respect to the upper-level decision variable (the side-payment decision variable x). This computation is nontrivial.  One of the major contributions in this paper (Proposition 2) is an analytical expression of the gradient, with which we can carry out adaptive incentive design using a gradient estimator without the knowledge about the follower’s reward function.
>
> Ref:
>
> [1] B. Liu, M. Ye, S. Wright, P. Stone, and Q. Liu, “BOME! bilevel optimization made easy: a simple first-order approach,” in Proceedings of the 36th International Conference on Neural Information Processing Systems, in NIPS ’22. Red Hook, NY, USA, pp. 17248–17262.
>
> [2] R. Liu, J. Gao, J. Zhang, D. Meng and Z. Lin, "Investigating Bi-Level Optimization for Learning and Vision From a Unified Perspective: A Survey and Beyond," in IEEE Transactions on Pattern Analysis and Machine Intelligence, vol. 44, no. 12, pp. 10045-10067, 1 Dec. 2022
>
> 2.  "The experiments part is not solid enough to verify the effectiveness of this method, without comparison of other state-of-art methods."
>
> Reply: All the existing methods for incentive design, model design, or reward shaping assume knowledge of the follower's rewards (see the introduction for more details). Therefore, they are not directly comparable to our method.
>
> Requested Changes:
> 1." Add comparison to other methods in the experiments part. It can also be compared with the ideal case where the follower’s reward function is known, to see the gap between the best possible result."
>
> Reply: We have compared two cases: incentive design with a known follower's reward (referred to as model-based) and incentive design with an unknown follower's reward (referred to as model-free), for both the small MDP example and the gridworld. However, it appears that the figure captions are missing, and only subfigure captions are included. We apologize for this oversight, which may have led to confusion in understanding the results. In both examples, we presented the convergence results of incentive design in the case of known rewards versus unknown rewards. We also demonstrated how different sample sizes affect convergence using stochastic hypergradient descent.
>
> 2. " What is the relationship between Section 4 Incentive-invariant reward shaping and the main results? Is section 4 proposing another way to estimate the follower’s reward function compared to the Monte Carlo policy evaluation proposed in section 3?"
>
> Reply: The main results of this paper are as follows: First, we show that the hypergradient for the bi-level optimization problem can be computed analytically when the leader knows the follower’s reward function and can compute the follower’s best response. Second, we demonstrate that this hypergradient can be estimated using sampled trajectories from the follower’s response, enabling adaptive incentive design without knowing the follower’s reward. Third, the proposed method for the case with unknown rewards can also be applied to a learned reward function. This last point relates to the contribution discussed in Section 4.
>
> The major question of incentivize-with-learned-reward is, “what guarantee do we have when we learn the reward function from trajectories and then apply incentive design to the learned reward function?”  This question arises because the learned reward function may not exactly match the follower’s original reward; it could be any reward function that is policy-invariant to it. In Section 4, we address this question by showing that the method will yield optimal incentive design for any learned reward function, as long as the learned reward is policy-invariant to the follower’s original reward. We will revise the draft to make the contribution in section 4 clear.

---

### Review · Reviewer_MbQe · 2024-11-24

**Summary Of Contributions:**

The paper addresses the problem of incentive design in a leader-follower setting, i.e. how can the leader influence the follower's policy in order to maximize their payoff. The particularity of the problem tackled is the fact that the leader does not know the follower's reward function.
After introducing the necessary background, the paper presents its main gradient based algorithm to find the optimal incentive design solely based on observations of the follower's behavior. It also provides some theoretical properties and guarantees on the algorithm, before showing empirical results on its behavior in simple environments such as gridworld.

**Audience:**

Yes

**Claims And Evidence:**

Yes

**Requested Changes:**

See weaknesses section.

**Strengths And Weaknesses:**

Strengths:
- Incentive design with no knowledge of the follower's reward function is an important yet underinvestigated problem
- Gradient based algorithm is an interesting solution that shows promising results
- Theoretical analysis gives some good mathematical grounding to the method

Weaknesses:
- Clarity: paper lacks clarity at times and it can be hard to understand what is the objective of each section - e.g. section 3, "Main result" does not say much, while Algorithm 2 is only introduced much later.
- Section 4 is not very novel. It basically corresponds to Inverse Reinforcement Learning, but no work or literature from the field has been cited nor discussed. "there can be multiple reward functions that are consistent with the data" is the reward ambiguity coming from Inverse Reinforcement Learning being ill-posed.
- Method is tested only in very simple if not toy settings. While these are good for understanding the behavior of the method, the paper lacks testing in a more complex setting.
- No code for reproducibility

---

> ### Author Response · Authors · 2024-12-04
> **Reply to Reviewer MbQe**
>
> Thanks for your valuable feedback! We address each weakness mentioned separately in a point-by-point manner.
>
> 1. Clarity: Thank you for the suggestions. We will revise the titles of each section to make them more informative and reflective of the content and modify the structure of the paper to make it easier to read and understand. Regarding the location of Algorithm 2, it is introduced after proving the correctness of the proposed method. It is noted that the main algorithm--Algorithm 1-- is introduced earlier and Algorithm 2 replaces the gradient computation in algorithm 1 using the gradient estimate from the follower’s trajectories.
>
> 2. Section 4 will be revised to include related references [1,2] on inverse reinforcement learning and its associated reward ambiguity problem [2,3] where multiple reward functions can equally explain the same expert behaviors. The main reason for including section 4 is to show that the proposed method of incentive design with a known reward function of the follower can be applied to a learned reward function, provided that the learned reward function and the original, unknown follower’s reward function are policy-invariant under reward shaping.
>
> 3.  Regarding the experiments, our theory and algorithm apply to any MDPs with finite state and action spaces, for which the grid world environment is a good representation and provide more interpretable results on the incentive design. If desired and time permitting, we may consider adding experiments with environments of various sizes to show how the computational time scales empirically.
>
> 4. Thank you for the comment. We will provide a git repo in the revision of this manuscript for reproducibility. The code has been made ready to be shared but due to the rule of rebuttal we cannot provide additional information at this stage.
>
> Refs.
> [1] A. M. Metelli, “Recent Advancements in Inverse Reinforcement Learning,” Proceedings of the AAAI Conference on Artificial Intelligence, vol. 38, no. 20, Art. no. 20, Mar. 2024, doi: 10.1609/aaai.v38i20.30296.
>
> [2] S. Arora and P. Doshi, “A survey of inverse reinforcement learning: Challenges, methods and progress,” Artificial Intelligence, vol. 297, p. 103500, Aug. 2021, doi: 10.1016/j.artint.2021.103500.
>
> [3] A. Baheri, “Understanding Reward Ambiguity Through Optimal Transport Theory in Inverse Reinforcement Learning,” Oct. 18, 2023, arXiv: arXiv:2310.12055. doi: 10.48550/arXiv.2310.12055.

---

### Review · Reviewer_soZq · 2025-01-02

**Summary Of Contributions:**

The paper addresses the problem of incentive design in Markov Decision Processes (MDPs) where a leader aims to influence a follower's behavior through reward modifications without knowing the follower's original reward function.

Key contributions include:

- Development of a gradient-based algorithm to compute optimal incentive designs without requiring knowledge of the follower's reward function
- Theoretical analysis proving convergence of the proposed method
- A method to estimate gradients from observed follower responses
- Analysis of conditions under which incentive designs remain optimal across different reward functions that are policy invariant

**Audience:**

Yes

**Broader Impact Concerns:**

No concerns on ethical implications.

**Claims And Evidence:**

Yes

**Requested Changes:**

Would be interested to see more discussions around the local vs. global optimiality of the proposed method.

**Strengths And Weaknesses:**

Strengths
1. The proposed method doesn’t require knowing follower’s reward function, which makes it quite general
2. Good theoretical foundations with convergence guarantees
3. Experimental validation on probabilistic transition systems and gridworld environments shows the effectiveness of the method

Weakness (not necesarily weakness, more towards future directions)
1. I’m not sure about the scalability of the method since it scales with state-action space size.
2. The experimental validation could be expanded to more complex scenarios

---

> ### Author Response · Authors · 2025-01-09
> **Reply to Reviewer soZq**
>
> We greatly appreciate the reviewer's comments and summary of the contribution. For the weakness and requested changes, we provide our response below:
> 1. Regarding the scalability: In the current manuscript, we have included a complexity analysis per gradient computation, right after Theorem 1. This analysis shows that the complexity is influenced by the state-action size of the underlying MDP and the number of side-payment decision variables. Similarly, classical complexity analyses of MDPs often address scalability concerning the size of state-action space. Because the method uses a gradient-based optimization and the objective function is nonconvex, it is difficult to estimate how many number of iterations required for the gradient-based algorithm to converge. Since the problem is solved offline and the per-step gradient computation is polynomial, this approach can be applied to many practical problems without significant scalability concerns.
>
> We also discussed how to reduce the computation by pre-selecting a subset of state-action pairs for allocating side-payments. This is one way to allow the method to scale for larger-size MDP incentive design.
>
> 2. We acknowledge the need for more complex scenarios and will try to include more results if time permits. The reason for using gridworld example is that the stochastic gridworlds are commonly used for MDP planning and design problems, and is considered as a benchmark case. More complex scenarios could consider changing different configurations of the side-payments and agent’s original reward/goal locations.
>
> 3. Requested Changes: Adding more discussion about the local vs. global optimality of the proposed method.
> Thank you for the suggestion! The objective function is non-concave and therefore the proposed gradient-based optimization method only ensures the convergence to a local optimal solution. In the revised manuscript, we show that for a simple follower’s MDP and some leader/follower reward functions, there exists multiple local optimal solutions and only one is global optimal solution (see Lemma 1 on page 5).

---

### Decision · Action_Editor_6fcr · 2025-03-02

**Recommendation:** Accept with minor revision

**Comment:**

The recommendation is to accept the paper subject to the following minor revisions that the authors promised in the rebuttal, but did not include in the latest version:

1. Add a discussion and references about related work in inverse RL.
2. Provide a link to the source code for reproducibility.
3. Improve the clarity by revising section titles to be more informative and revising the structure of the paper.

**Audience:**

This work will be of interest to the multiagent RL community.

**Claims And Evidence:**

The paper describes a new RL technique to incentivize a follower agent to execute actions that will be beneficial to the leader agent.  The paper claims that the proposed algorithm does not need to know the reward function of the follower.  A theoretical analysis supports the claim.  In addition, experiments in simple maze problems provide empirical evidence.  Overall, this represents a good contribution to the advancement of incentive design.

---

> ### Author Response · Authors · 2025-03-18
> **Response letter**
>
> Thank you for the editor and reviewers' suggestions. In response, we prepare the final submission with the following revisions:
>
> -. Add a discussion and references about related work in inverse RL.
> Reply: We added related work on inverse RL and clarify the discussion in section 4:
>
> " Our proposed adaptive incentive design requires no knowledge or learning of P2’s reward function. An alternative approach is to learn P2’s reward function from observed trajectories of P2, using inverse reinforcement learning methods (Abbeel & Ng, 2004; Ng & Russell, 2000; Ramachandran & Amir, 2007; Ziebart et al.,  2008), and then apply the incentive design with the learned P2’s reward case. For more information about inverse reinforcement learning algorithms, the readers are referred to the survey by Arora & Doshi (2021)."
> " Let’s refer the second method as reward learning-based incentive design. However, it is known that inverse reinforcement learning is ill-posed because there can be multiple reward functions that generate the same  optimal policy (Arora & Doshi, 2021). In that case, no amount of data can distinguish them. These reward functions are known as policy invariant."
>
> -. Provide a link to the source code for reproducibility.
>
> We added a link to the source code, on page 11, Section 5: Code: https://github.com/alexalvis/IncentiveFollower.
>
> -. Improve the clarity by revising section titles to be more informative and revising the structure of the paper.
> Reply: We have revised the section titles as follows to be more informative:
>
>  3. Adapting Incentive Design via Gradient Ascent
>
>  3.1 Computing the total gradient: The case with known P2's reward
>
>  3.2 Estimating the total gradient: The case with Unknown P2's reward